# Factors Predicting Type I Gastric Neuroendocrine Neoplasia Recurrence: A Single-Center Study

**DOI:** 10.3390/biomedicines11030828

**Published:** 2023-03-09

**Authors:** Mohammad Sheikh-Ahmad, Leonard Saiegh, Anan Shalata, Jacob Bejar, Hila Kreizman-Shefer, Majd F. Sirhan, Ibrahim Matter, Forat Swaid, Monica Laniado, Nama Mubariki, Tova Rainis, Ilana Rosenblatt, Ekaterina Yovanovich, Abed Agbarya

**Affiliations:** 1Institute of Endocrinology, Bnai Zion Medical Center, 47 Golomb St., Haifa 31048, Israel; leonard.saiegh@b-zion.org.il (L.S.); anan.shalata@b-zion.org.il (A.S.); ilana.rosenblat@b-zion.org.il (I.R.); katya.rogachevsky@b-zion.org.il (E.Y.); 2The Ruth and Bruce Rappaport Faculty of Medicine, Technion, Haifa 31096, Israel; ibrahim.matter@b-zion.org.il (I.M.); forat.swaid@b-zion.org.il (F.S.); monica.laniado@b-zion.org.il (M.L.); tova.rainis@b-zion.org.il (T.R.); abed.agbarya@b-zion.org.il (A.A.); 3Department of Pathology, Bnai Zion Medical Center, 47 Golomb St., Haifa 31048, Israel; jacob.bejar@b-zion.org.il (J.B.); hila.shefer@b-zion.org.il (H.K.-S.); majd.sirhan@b-zion.org.il (M.F.S.); 4Department of Surgery, Bnai Zion Medical Center, 47 Golomb St., Haifa 31048, Israel; 5Division of Gastroenterology, Bnai Zion Medical Center, 47 Golomb St., Haifa 31048, Israel; naama.mabariki@b-zion.org.il; 6Department of Oncology, Bnai Zion Medical Center, 47 Golomb St., Haifa 31048, Israel

**Keywords:** type I gastric neuroendocrine neoplasm, mitotic count, Ki-67 index, chromogranin A, gastrin, recurrence

## Abstract

Type I gastric neuroendocrine neoplasms (gNENs) are associated with atrophic gastritis and have a high recurrence rate, which means frequent endoscopies are required. The objective of this study was to identify factors predicting the local recurrence of type I gNENs. The clinical course and the pathological and biochemical data of patients with type I gNENs treated at Bnai Zion Medical Center between 2006 and 2022 were analyzed retrospectively. Twenty-seven type I gNENs were evaluated. The follow-up period was 41 months (range: 11–288 months). Recurrence of the tumor occurred in 13/27 (48%) patients after 35 months (median (M), interquartile range (IQR): 21–67.5). Serum gastrin levels were significantly higher in patients with recurrent disease versus patients with non-recurrent disease (788 vs. 394 ng/L; *p* = 0.047), while the Ki-67 index was significantly lower in patients with recurrent disease versus patients with non-recurrent disease (1% vs. 3.5%; *p* = 0.035). Tumor size, mitotic count, and serum chromogranin A levels did not correlate with recurrence. The present study emphasizes the role of gastrin in the pathogenesis of gNEN recurrence and highlights the debate regarding the ability of the Ki-67 index to predict the clinical course of this disease.

## 1. Introduction

Gastric neuroendocrine neoplasm (gNEN) is a tumor derived from enterochromaffin-like (ECL) cells localized in the gastric mucosa. Three subtypes of gNENs have been identified: type I gNEN (70–80%) is associated with chronic atrophic gastritis and hypergastrinemia; type II gNEN (5–10%) is associated with gastrinomas in Zollinger–Ellison syndrome and multiple endocrine neoplasia type I; and type III gNEN (15–20%) is an aggressive sporadic tumor arising in otherwise normal gastric mucosa without hypergastrinemia [1].

Neoplastic changes in type I gNENs are always associated with an elevated concentration of serum gastrin, which exerts a trophic effect on ECL cells sequentially undergoing hyperplasia, dysplasia, and neoplasm formation [2]. Type I gNENs are non-functioning lesions, typically found during routine upper gastrointestinal endoscopies performed for dyspepsia or anemia. The incidence of gastric neoplasms has increased up to 15-fold over the past three decades, a period characterized by a major concurrent increase in the number of gastric endoscopies performed [3,4,5,6,7]. During this period, type I gNEN prevalence increased accordingly, from accounting for 4% to 22.8% of all gastrointestinal neuroendocrine tumors [6,8].

Type I gNENs frequently present as multiple polyps, usually less than 1 cm in diameter, localized in the gastric corpus fundus. Grade 1 denotes a well-differentiated tumor, with a Ki-67 labeling index <3% and a mitotic index <2 per 10 high-power fields (10HPF); grade 2 denotes a moderately differentiated tumor, with a Ki-67 labeling index of 3–20% or a mitotic index of 2–20 per 10HPF [1]. These tumors have good prognoses: a systematic review of type I gNENs confirmed the indolent course of this tumor type, with very low disease-specific mortality [9]. Only five tumor-related deaths were reported in more than one thousand patients, all of whom had exceptional disease characteristics, such as large tumor size, grade 3 histology, or metastatic disease at presentation [9]. Moreover, despite the increase in type I gNEN detection, in everyday clinical practice, we do not encounter an increased prevalence of metastatic cases [8,10].

Therapeutic strategies for the treatment of type I gNENs (such as endoscopic surveillance or surgery) are based on risk stratification according to tumor size, number of lesions, disease stage, and tumor grade [3]. The European Neuroendocrine Tumor Society (ENETS) guidelines for the management of patients with type I gNENs suggest endoscopic lesion resection with periodic endoscopic surveillance. Gastrectomy is reserved for cases with a large tumor, tumor invasion beyond the submucosa, multiple lesions, and/or with metastasis [11].

As hypergastrinemia is continuous in chronic atrophic gastritis and plays a key role in the pathogenesis of type I gNENs, local recurrence should be expected. Several studies demonstrated a wide range of recurrence rates, from 5% to 65%, showing various parameters that correlated with recurrence, with no consistent conclusions [12,13,14,15,16,17]. Therefore, the present study aimed to evaluate pathological and biochemical parameters in correlation with the local recurrence of type I gNENs in patients treated at Bnai Zion Medical Center.

## 2. Materials and Methods

### 2.1. Study Population and Design

This single-center retrospective study enrolled patients diagnosed with type I gNEN at Bnai Zion Medical Center, Haifa, Israel, between the years 2006 and 2022. The study was conducted according to the guidelines of the Declaration of Helsinki and approved by the Institutional Review Board of Bnai Zion Medical Center.

The diagnosis of a type I gNEN was established based on the histopathological features of the gastric lesion excised during gastric endoscopy, along with atrophic gastritis demonstrated on the histopathology of random stomach mucosa biopsies (Figure 1).

The inclusion criteria were patients aged ≥18 years with a diagnosis of gNEN and atrophic gastritis via histopathology. Exclusion criteria were patients diagnosed with gastric neuroendocrine carcinoma or type II and type III tumors, or patients without a comprehensive medical history or complete information regarding the tumor excision protocol and endoscopic data.

### 2.2. Clinical Evaluation

Lesions less than 10 mm in diameter were resected endoscopically with biopsy forceps or using the cold snare technique. For ≥10 mm lesions, an endoscopic ultrasound was performed to determine the presence of pathological lymph nodes and local tumor invasion. Polypectomy was performed using either endoscopic mucosal resection or endoscopic submucosal dissection.

The tumors were graded according to the 2019 World Health Organization classification and grading system for neuroendocrine neoplasms of the digestive system [1].

The patients were followed up with a gastroscopy 6 months later. A gastroscopy showing no evidence of tumor recurrence warranted a repeated gastroscopy one year later.

A partial gastrectomy was considered in cases of high-grade tumors, metastatic disease, invasion of the tumor to muscularis propria, or in cases of tumor diameters ≥ 2 cm. A discussion regarding the necessity for surgery was held by a tumor board forum that included surgeons, gastroenterologists, endocrinologists, and oncologists, considering surgical risks, benefits, and patients’ preferences.

Data were collected from the patients’ medical records. The variables analyzed were age, gender, tumor size (tumor largest diameter via histopathology), mitotic count, Ki-67 index, serum gastrin level at diagnosis, serum chromogranin A level at diagnosis, resection approach (endoscopic or surgical), depth of tumor invasiveness (tumor invading the lamina propria, submucosa, or muscularis propria), gallium 68 DOTATATE PET-CT results, helicobacter pylori test in pathology samples, somatostatin analog treatment, and time elapsed until the first recurrence.

### 2.3. Tissue Processing and Immunohistochemistry

Biopsy samples were fixed in 4% paraformaldehyde, dehydrated in ascending alcohol concentrations, and embedded in paraffin. Formalin-fixed, paraffin-embedded tissues were sequentially cut into 3 µm sections and mounted on positively coated slides. The histopathological evaluation of hematoxylin–eosin-stained sections was performed, and the mitotic count was calculated by counting mitotic events per 10HPF. Immunostains were performed using an automated stainer (Benchmark Ultra; Ventana Systems, Phoenix, AZ, USA). Antigen retrieval in Tris-based buffer (36 min at 95–100 °C) was followed by 32-min primary antibody incubation for mouse anti-human Ki-67 (clone MIB-1, Dako, Glostrup, Denmark) and 24-min incubation for mouse anti-human Chromogranin A (clone LK2H10, Merk KGaA, Darmstadt, Germany). An UltraView DAB detection kit (760–500, Ventana Systems, Phoenix, AZ, USA) was used in the detection reaction according to the manufacturer-recommended protocol. The hematoxylin counterstains were used for color development. The expression of the Ki-67 index protein was determined by calculating the ratio of positively stained nuclei among 100 tumor cells.

### 2.4. Data Analysis

The statistical analysis of the data was performed using SPSS version 21 software (IBM, Chicago, IL, USA). Descriptive statistics for all the parameters included in this study were presented in terms of the median (M), interquartile range (IQR) percentiles, and percentages. The normal distribution of the quantitative parameters was determined by the Kolmogorov–Smirnov test, and the Mann–Whitney U non-parametric test was employed to determine differences between groups. Fisher’s exact test was used to examine categorical parameters, and Pearson correlation was used to test the relation between the level of serum gastrin and the level of chromogranin A. A *p*-value < 0.05 was considered statistically significant.

## 3. Results

Out of 36 patients who met the inclusion criteria, 27 patients were enrolled, and 9 patients were excluded (1 due to a diagnosis of gastric neuroendocrine carcinoma and 8 due to incomplete medical data).

The median follow-up period was 41 months (range: 11–288 months). Local recurrence of the tumor occurred in 13/27 (48.1%) patients, with a median time to recurrence of 35 months (M, IQR: 21–67.5) (Table 1). In total, 9 out of the 27 tumors were classified as grade 2; 5/9 had mitotic counts ≥ 2, while 8/9 had Ki-67 labeling indexes ≥3%. Partial gastrectomy (wedge resection) was performed in six cases, including five out of the seven cases with tumor diameters ≥10 mm. Gallium 68 DOTATATE PET-CT imaging was performed in all patients treated with wedge resection of the stomach and in seven patients with local disease, and only one case with lymph-node metastasis was revealed (15 mm neoplasm, grade 1, invading submucosa, mitotic index of 0, Ki-67 less than 1%, and 6 out of 17 lymph nodes resected were positive for metastasis, Figure 2). Four patients were treated with a somatostatin analog: one patient with a locally recurrent disease, one patient with an extremely high gastrin level of 2893 ng/mL, one patient with a 2.5 cm tumor diameter, and one patient with a metastatic disease. At the time of diagnosis, only two patients were treated with proton-pump inhibitors (PPIs). They had high levels of serum gastrin (1400 and 1658 ng/L) and did not have recurrent disease. None of the study patients remained on PPI treatment during follow-up.

The following variables did not correlate with recurrent disease: age, gender, mitotic count, chromogranin A level, size of the tumor, and resection approach (Table 2).

Serum gastrin levels at presentation were significantly higher in patients with a recurrent disease versus patients with a non-recurrent disease, while the Ki-67 index was significantly lower in patients with a recurrent disease versus patients with a non-recurrent disease (Table 2).

Twenty-two gNENs had mitotic counts of 0–1. Mitotic counts of 0–1 versus ≥2 showed no correlation with the gastrin level (*p* = 0.38), chromogranin A level (*p* = 0.37), or the invasiveness of the tumor (*p* = 0.56).

In 19 tumors, the Ki-67 index was <3%. A Ki-67 index < 3% versus ≥3% showed no correlation with the gastrin level (*p* = 0.15), chromogranin A level (*p* = 0.35), or the invasiveness of the tumor (*p* = 0.56).

The diameter of the tumor in seven (25.9%) patients was ≥10 mm. Tumor diameter showed no correlation with the gastrin level (*p* = 0.29), chromogranin A level (*p* = 0.31), or the invasiveness of the tumor (*p* = 1).

The Pearson correlation test revealed a positive and significant linear correlation between the Ki-67 index and chromogranin A level, r = 0.626, *p* < 0.001. No significant (*p* = 0.49) correlation was found between the level of gastrin and the level of chromogranin A.

The anti-parietal cell antibody was positive in 12/12 patients tested. Helicobacter pylori results were available for 22 patients, of whom only 2 were positive.

## 4. Discussion

The present study shows that high gastrin levels and low Ki-67 indexes are significantly correlated with type I gNEN local recurrence.

The current findings showed that tumor size did not correlate with the recurrence rate. Tumor size is one of the most important factors affecting patient management according to ENETS international guidelines. The current guidelines suggest an aggressive approach toward large tumors > 10 mm, which should be resected via surgery, vs. tumors less than 10 mm, which can be treated using an endoscopic approach [11]. However, no large prospective studies have been carried out to confirm these recommendations. Some studies showed that tumors sized between 10 and 20 mm could be followed up endoscopically. Experiences in the management of type I gNENs at Mount Sinai demonstrated that the outcomes of all tumors were good in every therapy modality that was applied (such as polypectomy, somatostatin analog treatment, or surgical resection) [18]. Consequently, they concluded that the decision between tumor resection and watchful surveillance should mainly depend on the risk of resection, especially if multiple resections are required. A study by Panzuto et al. found that a tumor size larger than 10 mm was associated with poor outcomes (local lymph nodes and angioinvasion via histology), while it was not associated with either five-year survival rates or with local recurrence [19].

In this study, a significant correlation was found between high serum gastrin levels and local recurrence risk. However, previous publications have shown mixed results regarding this issue. A study that included 66 patients with gNENs, subdivided into long-term (n = 38) and short-term (n = 28) PPI users, demonstrated that after the removal of their initial tumors, 5/38 in the former group experienced recurrence, while no one in the latter group experienced recurrence [20]. As PPIs are well known to cause gastrin elevation, the longer use of PPIs may result in higher gastrin levels, and this may indicate an association between higher gastrin levels and the risk of tumor recurrence. Moreover, and in accordance with the results of the current study, a previous study that recruited 114 patients with type I gNENs who were followed up annually showed that high levels of serum gastrin were related to local recurrence [13]. On the other hand, various reports did not find that correlation to be true. In a study that reviewed the medical records of 103 patients who underwent endoscopic resections of type I gNENs, during a median follow-up period of 63 months, local recurrence rates were found to be 6.5% and 2.4% in the endoscopic mucosal resection and endoscopic submucosal dissection groups, respectively, without correlation with gastrin levels [14]. Furthermore, the present study demonstrated that type I gNEN recurrence does not correlate with tumor size, depth of invasion, or tumor grade [14]. In their study on 97 patients with type I gNENs, Campana et al. showed that 26.2% of the patients had disease recurrence after endoscopic resection and 26.3% after somatostatin analog treatment [2]. They found no correlation between disease recurrence and gastrin level, Ki-67 index, gender, type of therapy (medical therapy versus endoscopic resection), number of neoplastic lesions (less than or more than five lesions), or tumor grade (grade 1 versus grade 2) [2].

More data from previous studies regarding recurrence rates and factors associated with the recurrence of type I gNENs are shown in Table 3.

The Ki-67 index is considered a crucial marker in predicting the risk of recurrence and death in patients with gastro-entero-pancreatic neuroendocrine neoplasms [21]. The finding of the current study, that a high Ki-67 index did not correlate with gNEN recurrence, might emphasize the need to re-evaluate the role of the Ki-67 index in type I gNENs. Previous studies also failed to find a correlation between the Ki-67 index and the recurrence rates of type I gNENs [2,13,18]. Moreover, conflicting results have been reported regarding the role of the Ki-67 index in predicting disease prognosis. In a retrospective study of 20 patients with metastatic type I gNENs, the Ki-67 index widely ranged from 1% to 20%, and 11/20 of these patients had Ki-67 less than 3% [22]. Ki-67 labeling index values ranging from 0.1% to 15% were also reported in another large series of 111 patients with type I gNENs, who showed excellent long-term survival with no tumor-related death, regardless of the Ki-67 labeling index value [23].

As the Ki-67 index and mitotic count may be unmatched in the same tumor, the WHO and ENETS recommend applying both approaches to grade tumors reliably, and in cases of discrepancy, the higher value determines the grade [11]. However, there is an ongoing debate regarding what is more reliable, the mitotic count or the Ki-67 labeling index. The discrepancy between the mitotic count and the Ki-67 labeling index in the evaluation of tumor grade could be attributed to the fact that these parameters describe different phases in the cell cycle. Ki-67 protein is observed in proliferating cells that do not present mitotic features (mid-G1 through S and G2 phases), while mitosis is a separate phase that occurs during cell division (M phase) and is a shorter event in the course of the cell cycle [24]. The advantage of mitotic count is that it is performed in the routine processing of hematoxylin-stained tissue slides, while the Ki-67 labeling index requires immunohistochemical studies. Moreover, a pivotal drawback in the evaluation of the Ki-67 index is the unintentional counting of non-neoplastic cells that may be in a proliferative state within the tumor sample. Gastrointestinal neuroendocrine tumors were found to include several types of cells expressing Ki-67, such as intratumoral endothelial cells, background epithelium (e.g., glands or crypts), and lymphocytes [25] (Figure 2B). Nevertheless, the Ki-67 labeling index defines the percentage of cells expressing the protein and therefore enables the assessment of small-size biopsies [26].

The use of the Ki-67 index and/or mitotic count according to the WHO and ENETS grading systems was validated for foregut and pancreatic NENs by several studies, and the power of the Ki-67 index and/or mitotic count to determine prognosis has been suitably confirmed [27,28]. However, Ki-67 expression does not reliably separate typical from atypical lung carcinoids [29], and to the best of our knowledge, a study providing a validation of the prognostic role of Ki-67 in gNENs has not been published yet. It has been recommended that a high Ki-67 index value (≥3%) should indicate surgical resection [3]; however, these recommendations may be questionable in light of the published data.

The present findings, alongside previously published data, highlight the heterogeneous behavior of type I gNENs and the confusing information regarding risk stratification and the choice of the appropriate mode of treatment. Consequently, decisions regarding the management of type I gNEN patients are not straightforward and should involve several medical disciplines. A recent trial showed that a multidisciplinary team board changed the treatment decision in 50% of patients with neuroendocrine tumors arising from different sites [30]. The team involved in the management of type I gNEN patients should at least include an experienced pathologist for accurate grading, a gastroenterologist to consider the benefit and risk of repeated local resections, a surgeon to consider the risk of surgical procedures, an endocrinologist to consider medical treatment, and an oncologist.

A major benefit of our study is the demonstration of a positive correlation between serum levels of gastrin and type I gNEN tumor recurrence. Accordingly, patients with very high serum gastrin levels may be suitable for closer endoscopic surveillance than those with mildly evaluated levels. In the same manner, and to prevent local recurrence, clinicians may consider the use of somatostatin analogs for the inhibition of gastrin secretion. It was previously demonstrated that treatment with somatostatin analogs resulted in local recurrence rates that were comparable to endoscopic resection [2]. Therefore, treatment with these drugs may be considered in patients with multiple gastric lesions or in cases with several recurrences when gastrin levels are extremely high. In addition, small preliminary studies demonstrated that type I gNENs showed complete regression during treatment with the gastrin receptor antagonist “netazepide” [31]; thus, this treatment might also be considered suitable in the future. These treatment strategies may spare patients from the possible adverse consequences of repeated endoscopic resections or unfavorable surgery. The current study also highlights the doubts regarding the value of the Ki-67 index in predicting the clinical course of this disease. As we showed that higher Ki-67 indexes did not correlate with local recurrence, we recommend that treatment-related decision making should not rely solely on the Ki-67 index, especially when considering aggressive treatment such as surgical resection.

The main limitation of our study is the low number of patients included. Another limitation is the difficulty in diagnosing a true local recurrence. In fact, some patients may have had invisible intramucosal gNENs at the time of the first diagnosis that were only visible at the time of the repeated endoscopy and therefore were falsely considered a recurrence. Moreover, removal using forceps or the core snare technique, which are used to treat some small polyps, might offer less complete resections, leaving part of the lesion in situ, which would be incorrectly considered a recurrence during follow-up.

## 5. Conclusions

This study shows that high gastrin levels at the time of diagnosis and a tumor with a low Ki-67 index were correlated with a high recurrence rate of type I gNENs. These results denote the possible role of gastrin-lowering drugs in the management of local tumor recurrence and highlight the debate regarding the role of the Ki-67 index in predicting the clinical course of the disease. More studies based on larger series are needed to evaluate these inspections. Moreover, this study shows that tumor size, mitotic count, and chromogranin A level are not correlated with recurrence.

## Figures and Tables

**Figure 1 biomedicines-11-00828-f001:**
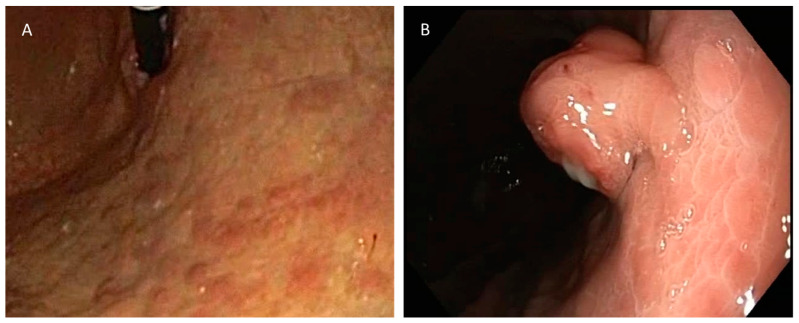
(**A**) Typical appearance of atrophic gastritis on endoscopy. (**B**) Gastric polyp, diagnosed via biopsy as gNEN.

**Figure 2 biomedicines-11-00828-f002:**
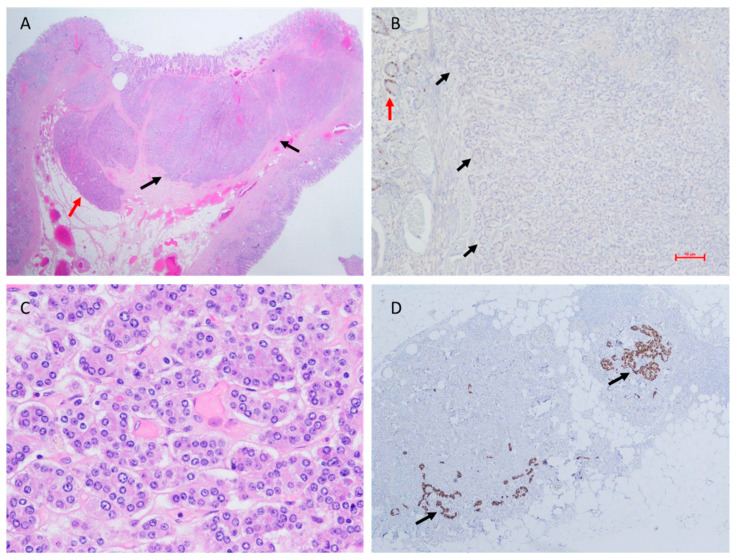
Histopathology of a metastatic type I gNEN, surgically resected. (**A**) Hematoxylin and eosin stain of the neoplasm (black arrows) showing infiltration of submucosa of the stomach wall (red arrow) (original magnification ×10). (**B**) Ki-67 index < 1% in the neoplasm (black arrows = neoplasm margins); positive Ki-67 immunohistochemical staining of a non-neoplastic gastric gland (red arrow) (original magnification ×100). (**C**) Mitotic count = 0 (original magnification ×400). (**D**) Lymph node metastases adjacent to the peri-gastric fat tissue (chromogranin A immunohistochemical staining, black arrows) (original magnification ×40).

**Table 1 biomedicines-11-00828-t001:** Demographic and clinical characteristics of type I gNEN ^1^ patients (n = 27).

Characteristic	Median (IQR) ^1^	Range (min.–max.)
Age at diagnosis (years)	63 (52–68)	35–75
Tumor size (mm)	7 (5–13)	2–30
Mitotic count (IQR, SD ^1^)	0 (0–1, ±1.65)	0–8
Ki-67 index	1% (1–5%)	0–20%
Gastrin ^2^ (ng/L)	598 (350–1100)	107–2893
Chromogranin A ^3^ (ng/mL)	299 (171–509)	42–4820
	n (%)	
Gender		
Female	19 (70.4)	
Male	8 (29.8)	
Grade		
1	18 (66.7)	
2	9 (33.3)	
Invasiveness of tumor		
Tumor invading lamina propria or submucosa	23 (85.2)	
Tumor invading the muscularis propria	4 (14.8)	
Procedure		
Endoscopic US ^1^ performed	8 (29.6)	
Endoscopic resection	21 (77.8)	
Surgical resection	6 (22.2)	
Metastatic disease	1 (3.7)	
Recurrences	13 (48.15)	

^1^ Abbreviations: IQR, interquartile range; gNEN, gastric neuroendocrine neoplasm; SD: standard deviation; US, ultrasound. ^2^ normal < 115 ng/L. ^3^ normal < 98.1 ng/mL.

**Table 2 biomedicines-11-00828-t002:** Comparison of clinical, biochemical, and histopathological variables between recurrent and non-recurrent type I gNEN.

	No Recurrence (n = 14)	Recurrence (n = 13)	*p*
Age (years) (mean ± SD)	62.2 ± 11.6	57.0 ± 11.3	*p* = 0.35
Gender, n (%)			*p* = 0.68
Female	9 (64.3%)	10 (76.9%)	
Male	5 (35.7%)	3 (23.1%)	
Mitotic count, n (%)			*p* = 0.21
0	8 (57%)	11 (85%)	
≥1	6 (43%)	2 (15%)	
0–1	10 (71%)	12 (92%)	*p* = 0.33
≥2	4 (29%)	1 (8%)	
Ki-67 index, n (%)			*p* = 0.035
<3%	7 (50%)	12 (92%)	
≥3%	7 (50%)	1 (8%)	
Ki-67 index *	3.5% [1–8.5]	1% [1–1]	
Gastrin level * (ng/L)	394 [195–925]	788 [569–1000]	*p* = 0.047
Chromogranin A level * (ng/mL)	422 [197–633]	258 [137–494]	*p* = 0.34
Size of tumor (mm), n (%)			*p* = 0.38
<10	9 (64%)	11 (85%)	
≥10	5 (34%)	2 (15%)	
Size of tumor *, (mm)	8 [4.7–21.2]	7 [5–8]	*p* = 0.26
Invasiveness of tumor			
Tumor invading lamina propria or submucosa	11 (78.6%)	12 (92.3%)	*p* = 1.00
Tumor invading the muscularis propria	3 (21.4%)	1 (7.7%)	
Resection approach			*p* = 0.17
Endoscopy	9 (64%)	12 (92%)	
Surgery	5 (36%)	1 (8%)	
Follow up * (months)	41.5 [24–73.75]	41 [24–70.5]	*p* = 0.94

Data presented as n, frequencies, number of patients; and percent of total in parenthesis (%). * Median with 25–75% (IQR).

**Table 3 biomedicines-11-00828-t003:** Studies reporting type I gNEN recurrence rates.

Study	Recurrence Rate	Follow Up (Median)	Factors Correlated with Recurrence
Merola et al., 2012 [12]	21/33 (63.6%)	46 months	No risk factors were identified
Daskalakis et al., 2019 [13]	44/84 (52%)	45 months	High serum gastrin levels
Noh et al., 2021 [14]	5/103 (4.9%)	63 months	Incomplete endoscopic resection
Esposito et al., 2022 [15]	37/65 (56.9%)	48 months	Presence of multiple gNENs
Hanna et al., 2021 [16]	48/74 (64.9%)	63.7 months	Lesions 5 mm or larger
Tsolakis et al., 2019 [17]	Meta analysis of 8 studies: 75/422 (17.8%)	47–87 months	Non-surgical resection

## Data Availability

The data presented in this study are available on request from the corresponding author.

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
