# Peer review of "Factors Predicting Type I Gastric Neuroendocrine Neoplasia Recurrence: A Single-Center Study"

_biomedicines, 2023, doi:10.3390/biomedicines11030828_

Round 1
Reviewer 1 Report
The purpose of the study by Sheikh-Ahmad was to identify factors predictive of local recurrence of type I gastric neuroendocrine neoplasms (gNENs). 27 type I gNENs were evaluated over a 41-month follow-up period. Tumor recurrence occurred in 48% of patients at 35 months. Serum gastrin levels were found to be significantly higher in patients with recurrent disease compared to those with non-recurrent disease. The Ki-67 index was significantly lower in patients with recurrent disease compared to patients without recurrent disease. Tumor size, mitotic count and serum chromogranin A levels did not correlate with recurrence. The results of this study suggest that gastrin may play a role in the pathogenesis of gNEN recurrence and the ability of Ki-67 index to predict the clinical course of this disease is still uncertain.
In principle, I find the data interesting and relevant for the readership, especially those dealing with neuroendocrine tumors.
However, there are a few points that the authors should revise:
1. How do the authors see the relevance of their study results to practice? Should gastrin concentrations and Ki-67 be included in decision making for more aggressive therapy? This should be further discussed based on their own data and the literature, and the advantages and disadvantages of including these parameters should be described.
2. P-values should be consistently italicized in the tables.
3. Minor spelling errors are repeated throughout the text.
Author Response
Dear Reviewer,
We thank you for your valuable comments as these helped us to improve the manuscript.
Reply to comment 1:
We added to the article a discussion regarding this point:
“A major benefit of our study is the demonstration of a positive correlation between serum levels of gastrin and type I gNEN tumor recurrence. Accordingly, patients with very high serum gastrin levels may be suitable for closer endoscopic surveillance than those with mildly evaluated levels. In the same manner, and to prevent local recurrence, clinicians may consider the use of somatostatin analogs for the inhibition of gastrin secretion. It was previously demonstrated that treatment with somatostatin analogs resulted in local recurrence rates that were comparable to endoscopic resection [2]. Therefore, treatment with these drugs may be considered in patients with multiple gastric lesions or in cases with several recurrences when gastrin levels are extremely high. In addition, small preliminary studies demonstrated that type I gNENs showed complete regression during treatment with the gastrin receptor antagonist "netazepide"[31]; thus, this treatment might also be considered suitable in the future. These treatment strategies may spare patients from the possible adverse consequences of repeated endoscopic resections or unfavorable surgery. The current study also highlights the doubts regarding the value of the Ki-67 index in predicting the clinical course of this disease. As we showed that higher Ki-67 indexes did not correlate with local recurrence, we recommend that treatment-related decision making should not rely solely on Ki-67 index, especially when considering aggressive treatment such as surgical resection.”
Reply to comment 2:
P-values were italicized in the tables.
Reply to comment 3:
Language revision was made for the whole text by MDPI English editing service.
Reviewer 2 Report
The idea of the study could be of interest, however the theme has been vastly researched before on lager patient cohorts and without great statistically significant results. Given these considerations, here are some suggestions that could enhance the quality of the manuscript:
1. In the Introduction section, the paragraph describing the histopathological classification of type I gNEN (“Grade 1 denotes a well differentiated tumor…”) should be reformulated and included in the one above, as there is no need for a separate paragraph on this matter.
2. The patient inclusion criteria need to be reformulated and enhanced, for example, adding the request for complete information on the tumor excision protocol, patient medical history etc.
3. Please specify the complete name of the radiotracer used for PET/CT (Ga68-DOTATATE/DOATATOC etc.)
4. I suggest adding all the coefficients resulted from Pearson Correlation tests in a separate table, as they are hard to follow in the text.
5. Table 2 should be renamed, as it describes the findings between the two groups and not the correlations between the analyzed parameters.
6. The authors mention testing the anti-parietal cell antibodies in 12 patients. Did the positivity of this test correlate with the presence of recurrent disease or other parameters?
7. The authors stated that high serum gastrin levels correlated with disease progression, however they mention in the Discussion section that PPI treatment in these type of patients can lead to high gastrin levels. The information regarding whether the patients included in this study were or not on PPI treatment is missing in the text. I think that this is an important thing to specify in the manuscript, as the authors, unlike other research groups, found that gastrin levels correlate to the risk of recurrence. Moreover, if the patients followed treatment with PPI, I believe it should also be included in the study limitations.
8. Please do an English check-up, as some phrases might be better reformulated and some grammar errors need to be reformulated (such as “local tumor invasion” in the first paragraph of section 2.2)
Author Response
Dear Reviewer,
We thank you for your valuable comments as these helped us to improve the manuscript.
Reply to comment 1:
We made these changes to the paragraph.
Reply to comment 2:
exclusion criteria were modified: "Exclusion criteria were patients diagnosed with gastric neuroendocrine carcinoma or type II and type III tumors; or patients without comprehensive medical history or complete information regarding the tumor excision protocol and endoscopic data."
Reply to comment 3:
The full name is gallium 68 DOTATATE PET-CT. We added it to the text.
Reply to comment 4:
The only significant result in the Pearson Correlation test was between Ki-67 index and chromogranin A level. Other correlations were not significant, so we chose not mention the "r" of these insignificant correlations.
Quoting from results:
The Pearson correlation test revealed a positive and significant linear correlation between the Ki-67 index and chromogranin A level, r=0.626, p<0.001. No significant (p=0.49) correlation was found between the level of gastrin and the level of chromogranin A.
Reply to comment 5:
Table 2 was renamed: Table 2. Comparison of clinical, biochemical, and histopathological variables between recurrent and non-recurrent type I gNEN.
Reply to comment 6:
All patients tested for anti-parietal cell antibody were positive, so we can not come with conclusions regarding correlations.
Reply to comment 7:
We added the relevant data to results: “At the time of diagnosis, only two patients were treated with proton-pump inhibitors (PPIs). They had high levels of serum gastrin (1400 and 1658 ng/l) and did not have recurrent disease. None of the study patients remained on PPI treatment during follow-up.”.
Reply to comment 8:
Language revision was made for the whole text by MDPI English editing service
Round 2
Reviewer 1 Report
The authors addressed all my comments adequately.
Reviewer 2 Report
Thank you for your prompt answer to my suggestions. The authors have performed the required changes accordingly, however there are still problems that need to be considered, such as the difficulties encountered when diagnosing true local recurrence, as stated as a limitation by the authors. Even the low number of parameters that had statistically significant correlations might represent an issue when analysing the scientific soundness of the paper. I still believe that the theme has been vastly researched before on lager patient cohorts without producing great statistically significant results, and the statements the authors made have been demonstrated before, but, if the Academic Editor agrees with this matter, the paper can be published in the current form.